# What Works for Controlling Meningitis Outbreaks: A Case Study from China

**DOI:** 10.3390/vaccines11121762

**Published:** 2023-11-27

**Authors:** Xiyu Zhang, Chunting Liu, Zongjun Yi, Linglu Zhao, Zhongju Li, Linhong Yao, Bufang Feng, Liping Rui, Bin Qu, Ming Liu, Fuqiang Cui

**Affiliations:** 1Vaccine Research Center, School of Public Health, Peking University, Beijing 100191, China; 2311110168@stu.pku.edu.cn; 2Beijing Key Laboratory of Toxicological Research and Risk Assessment for Food Safety, Beijing 100191, China; 3Center for Infectious Diseases and Policy Research & Global Health and Infectious Diseases Group, Peking University, Beijing 100191, China; 4Guizhou Center for Disease Control and Prevention, Guiyang 550004, China; lct2004lct@163.com (C.L.); zhaolinglu0707@163.com (L.Z.); rlp1201@163.com (L.R.); 5Zunyi Center for Disease Control and Prevention, Zunyi 564600, China; zyjkyzj886@163.com (Z.Y.); lzj_2023@163.com (Z.L.); 18585740343@163.com (B.Q.); 6Tongzi Center for Disease Control and Prevention, Zunyi 563200, China; ylh200310@163.com (L.Y.); 18685267073@163.com (B.F.)

**Keywords:** meningitis, outbreak, breakthrough

## Abstract

The meningococcal meningitis (MM) vaccine reduces the incidence of MM significantly; however, outbreaks still occur in communities with high vaccine coverage. We aimed to analyze the driving factors of infection from a community outbreak. A total of 266 children aged 9 to 15 years old from the three junior high schools of Tongzi county were identified. We documented infection cases using laboratory tests and analyzed attack rates, infection rates and risk factors for transmission. The index case in School A was identified, and the attack rate in School A was 0.03%. Children showed a significantly low infection rate of MenC in School A (13.2% vs. 19.5% in total children, *p* = 0.002), while exhibiting significantly high infection rates of MenA in School B (44.1% vs. 24.8% in total children, *p* < 0.001) and MenB in School C (11.1% vs. 4.1% in total children, *p* = 0.015). The infection rate of MenA for females (30.0%) was higher (*p* = 0.055) than for males (19.9%). In School A, 63.19% of children were vaccinated against MenC, while in School B the rate was 42.65% and in School C, it was 59.26%. Three male MenC infection cases were detected as breakthrough infection cases in addition to the index case. The findings suggest that the current full-course immunization has limited long-term effectiveness and is inefficient in preventing the transmission of MM among older children.

## 1. Introduction

Meningococcal meningitis (MM) is an acute contagious infectious disease and also one form of bacterial meningitis that may cause epidemics [1]. As a global public health issue, the estimated MM incidence in the 1980s was about 25,000 to 200,000 cases annually [2], with an annual incidence rate ranging from 0.07 to 12.6 cases per 100,000 population from 1950 to 2002 [3]. The incidence of MM reached its peak in the spring of 1967, with 403 cases per 100,000 population in China, whereas it was controlled at 0.09 cases per 100,000 population from 2005 to 2010, contributing to the government’s long-term efforts in vaccine development and immunization [4]. However, despite the small number of MM cases, the high mortality and sequelae of MM impose a heavy burden on families. Compared to the mortality rate of 0.30 per 100,000 population in the United States of America in 2019, the mortality in China reached 0.51 per 100,000 population according to the GBD 2019 study [5]. According to the National Notifiable Diseases Registry System data, MM is the leading cause of mortality and has the highest mortality incidence among contagious infectious diseases [4].

The introduction and utilization of conjugate meningococcal C vaccines in the UK and the USA, which follow a 2-, 4-, 6- and 12–15-month schedule, reduced N meningitidis serogroup C disease by over 90%. In addition, near-elimination of *Haemophilus* influenzae has been documented due to the introduction of conjugate Hib vaccines [6]. However, as one of the major public health challenges, its progress in reducing the burden is significantly lagging behind other vaccines for preventable infectious disease [7]. In addition, breakthrough infection, defined as the detection of Neisseria meningitidis carriers in throat swabs after full-course immunization with the corresponding meningitis serogroup vaccine, is another huge challenge for immunization plans. Breakthrough infections occur from time to time, and some meningitidis serogroups not included in the current national immunization program may still cause large-scale infections. Outbreaks of MM are especially more frequently detected in children. In day-care settings or schools, healthy children (aged 6–18 years old in China) can be infected under close contact with MM cases or asymptomatic carriers, leading to further outbreaks. Previous literature has reported similar cases being infected by the index cases and, finally, resulting in deaths [8,9,10]. One study pointed out that meningitis was either the second or third most important infectious syndrome [11]. Due to the severe symptoms and complex complications, the MM outbreak has caused an enormous epidemiological and economic burden at the individual, community and national levels [12,13,14].

There is a lack of records on the characteristics of recent meningitis infections among Chinese children. It must be noted that immunization plans and emerging pandemics may have changed the past infection pattern, which may further require updating immunization plans and prevention and control strategies. Cheng [15] suggests that the distribution of MM pathogens in China has undergone significant changes after COVID-19, and relevant monitoring needs to be strengthened. However, the existing literature provides limited evidence for the immunization plans’ reform. In addition, vaccine characteristics also matter. At present, there are five types of meningococcal meningitis vaccines in China, with MPSV-A and MPSV-AC included in the national immunization plan, and children are vaccinated free of charge at the sixth month, ninth month, third year and sixth year after birth, respectively. The remaining MPCV-AC, MPSV-ACYW135 and MPCV-ACYW135 need to be administered to eligible children at their own expense. Therefore, most infants and children are vaccinated with PSV. However, on one hand, nonconjugated pneumococcal polysaccharide vaccines do not elicit a protective immune response in children younger than 2 years [16], and on the other, only the PCV vaccine has a memory-enhancing immune effect after repeated vaccination. This prevailing condition may cause concerns about the long-term protection against meningitis.

In addition, the decline of antibodies induced by vaccination, immuno-deficient individuals and exposure to a higher viral inoculum may cause breakthrough infections, potentially impacting vaccination strategies. On the other hand, if breakthrough infections occur rarely or are mild and have a comparatively low probability of causing wide transmission, watchful waiting may be appropriate. On the other hand, if breakthrough infections are comparatively common, additional vaccine doses, changes in vaccine formulations or non-pharmaceutical interventions should be considered as a response. Limited evidence has focused more on the infection and incidents among children under five years old, ignoring the potential risk of outbreaks of infection among school-age children. In dense classrooms, infections are more likely to occur and can quickly form outbreaks among children in the short term. This characteristic may exacerbate the potential risk of infection among school-age children and further cause a huge burden of disease.

Exploring the current driving factors of the MM outbreak epidemic is necessary, especially under the breakthrough infection in the high-immunization-coverage era in China. In addition, it is also necessary to evaluate the impact of prevention and control measures during the outbreak, to provide evidence for the precise prevention and control of the subsequent MM outbreak. This study aims to investigate the outbreak of meningococcal infection among school-age children in Tongzi County, Guizhou Province, at the end of 2022, to provide the evidence for formulating the improved prevention and control strategies for the subsequent outbreak of MM in other settings.

## 2. Method

### 2.1. Index Case Identification

An index case refers to the case infected with pathogens during an infection outbreak that was first detected and reported. On 17 November 2022, at 10:00 a.m., the Guizhou Provincial Centers for Disease Prevent and Control (Guizhou CDC) received notification of a suspected case of MM from Zunyi City. The patient was a 13-year-old male resident student from School A in Liaoyuan Town. After feeling unwell on the afternoon of November 15, this patient returned from school to his home in the same town, where only his grandfather and grandmother lived with him. In the morning of November 16, the student’s face was cyanotic (with ecchymosis on his face), eyelids puffy and he felt weak. He took a temperature of 38 °C and was then sent to the emergency department of the local hospital at around 7 o’clock. In the afternoon on November 16, the patient was transferred to a higher-grade hospital and diagnosed as a “Suspected MM case” and was uploaded to the pandemic network at 8:38 on 17 November and further confirmed as meningitidis serogroup C by laboratory diagnosis. The main method is to use real-time PCR to detect Neisseria meningitidis species and specific nucleic acid fragments of common serogroups in blood samples, and to capture specific genes of Neisseria meningitidis species (CtrA gene and Group C serogroup-specific gene). The patient died on November 20 and was eventually defined as the index case.

After the confirmation of the index case, the local health and education departments took prompt action. First, the school strictly implemented morning and afternoon inspection and a daily reporting system. Second, they ordered all medical institutions and townships to carry out an active search for suspected cases, improve awareness of suspicious symptoms of MM, report suspicious cases and carry out isolation and treatment in a timely manner. An active search for additional MM cases was performed through symptom monitoring in schools, communities and the hospital information system. Specifically, six carriers were detected among the index case’s close contacts at the same school. These carriers were isolated and observed at home for 10 days, and no suspicious clinical symptoms found. Medical observation was also conducted on their family members of the index case and carriers above, as well as the school’s teachers and students, and no suspicious symptoms, such as fever, headache and vomiting, were found. There was no suspected MM case or subsequent incident in this outbreak infection. Finally, the local government encouraged residents aged under 18 to carry out doses of MenA plus MenC vaccine revaccination for those unvaccinated before. For the school where the index case was located, the immunization experience of students was verified through systematic verification combined with child vaccination certificates. For other residents aged over 24 months having not complete vaccination against MM in the past, residents were vaccinated with doses of the MenA plus MenC vaccine according to the immunization program. Residents can also choose ACYW135 vaccines (not included in current immunization program) as an alternative to vaccinate based on the principles of being informed, voluntary and self-funded.

### 2.2. Environmental Description and Case Exploration

The index case was a resident student in School A in Liaoyuan Town, Tongzi County. School A has 57 classes, 3215 students and 208 staff members, allowing students to attend as day students. Most of the students and teachers left School A on November 17th because of the COVID-19 epidemic; thereafter, the provincial-, municipal- and county-level CDC searched actively for suspected cases in Tongzi County, and no other case was detected in this process.

The class the index case was in has a total of 52 students, including 26 resident students and 26 day students. This classroom is in the middle of the corridor on the south side of the School A teaching building, with 11 other classrooms on the same floor. Two large windows are open on the corridor and four small windows are open on the back side. The distance between the seats in the classroom is normal. The canteen consists of a total of two floors with an independent entrance and exit channel. A staggered peak dining system is implemented. The index case’s dormitory contains 10 beds (5 high beds and 5 low beds) inside and 8 other students lived in this room with the index case.

### 2.3. Data Collection and Analysis

We investigated the implementation progress for planned MM vaccination in Tongzi County. At the same time, we conducted a survey on the immunization experience of the MM vaccine, meningococcal-carrying and serum antibody testing on children in three schools: the school where the case was (School A), another middle school in the same town as School A (School B) and a middle school in a town far away from the town School A is in (School C). School B has 3051 students and 167 teachers, and School C has 558 students and 45 teachers. This study adopted judgment sampling based on the professional knowledge of CDC investigators. A certain number of students aged 10–15 in the same dormitory, class, grade, school and the same age group from other schools with the index case were surveyed.

The immunization experience of children in this study was surveyed by checking vaccination certificates, vaccination cards and parents’ memories. A total of 266 children were surveyed, including 144 in School A, 68 in School B and 54 in School C. Under in-formed consent, signing an informed consent statement, surveyors were allowed to collect venous blood and detect meningitis antibodies from children. The blood of a total of 246 children was collected, including 124 in School A, 68 in School B and 54 in School C.

We defined the infected case as: (a) a Neisseria meningitidis carrier is detected on the throat swab; (b) the serum test shows positive antibodies against meningococcal meningitis, but the sample has no corresponding meningitis serogroup vaccination experience. This study tested the student samples in three schools except for the index cases, and further calculated the infection rate of every meningitidis serogroup and the attack rate of MenC. In addition to infection cases, we also analyzed the breakthrough infection case. This study collected the class, gender, resident, vaccination, specimen collection time and laboratory test results from three schools (A, B, C).

SPSS 27.0 was used for statistical analysis application. One-way ANOVA was used for mono-factor analysis to describe the differences between groupings of different characteristics and explore the relationships between infection and characteristics.

## 3. Result

In 3215 children in School A, one MenC case (attack rate in School A: 0.03%) was detected as attacked after breakthrough infection and listed as the index case. However, no other new secondary case was detected (attack rate: 0 in School B vs. 0 in School C). We then identified 266 children aged 9 to 15 years old from the three junior high schools of Tongzi county. To explore the spatial mode of infection, this study explored the differences among the infection cases and rates in the children in Schools A, B and C in Tongzi county (Table 1). In 144 children in School A, there was laboratory evidence of only MenA infection for 11 children (7.6%), only MenB infection for 2 children (1.4%), only MenC infection for 9 children (6.3%), only Men W for 2 children (1.4%), MenA plus MenC for 9 children (6.3%) and MenA plus MenB plus MenC for 1 child (0.7%). In addition, one child (0.7%) was detected infected with unclassified Neisseria meningitides. In 68 children in School B, there was laboratory evidence of only MenA infection for 8 children (11.8%), only MenB infection for 1 child (1.5%), only MenC infection for 1 child (1.5%), MenA plus MenC for 21 children (30.9%) and MenA plus MenB plus MenC for 1 child (1.5%). In 54 children in School C, there was laboratory evidence of only MenA infection for 6 children (11.1%), only MenB infection for 5 children (9.3%), only MenC infection for 1 child (1.9%), MenA plus MenC for 8 children (14.8%) and MenA plus MenB plus MenC for 1 child (1.9%).

### 3.1. Mono-Factor Analysis

To explore the mode of Neisseria meningitides infection, this study explored the differences among the infection cases and rates of Group A, B, C and W Neisseria meningitides (MenA, MenB, MenC, MenW). Table 2 further presented Neisseria meningitides infection cases and rates of the children with different characteristics. The results showed a significantly low infection rate of MenC in School A (13.2% vs. 19.5% in total children, *p* = 0.002), whereas there were significantly high infection rates of MenA in School B (44.1% vs. 24.8% in total children, *p* < 0.001) and MenB in School C (11.1% vs. 4.1% in total children, *p* = 0.015). In addition, the infection rate of MenA for females (30.0%) was higher (*p* = 0.055) than for males (19.9%).

### 3.2. Vaccination and Breakthrough Infection

The long-term implementation of the corresponding immunization program has been carried out by local government. As shown in Table 3, 99.29% of the children at the corresponding ages in Tongzi County were vaccinated with the first dose and 97.28% with the second dose, and 99.24% with the first dose and 98% with the second dose in Liaoyuan County (also the subregion of Tongzi County), respectively. A vaccination survey of 30 children aged 1 to 6 years at the site where the case lived showed that both MenA and MenA plus MenC vaccination rates were 100%.

The index case was vaccinated with one vaccine against MenA on 29 February 2012, and two doses of vaccines against MenA plus MenC on 27 September 2012 and 28 October 2015, respectively. In our study, 18.75% of children in School A were detected as vaccinated with only one dose of vaccine against MenA plus MenC, 61.11% vaccinated with two doses of vaccine against MenA plus MenC, 18.06% without a vaccination record of vaccine against MenA plus MenC, 2.08% vaccinated with the ACYW135 vaccine and 63.19% vaccinated with vaccines against MenC. In School B, 4.41% of children were detected as vaccinated with only one dose of vaccine against MenA plus MenC, 42.65% vaccinated with two doses of vaccine against MenA plus MenC, 52.94% without a vaccination record of vaccine against MenA plus MenC and 42.65% vaccinated with vaccines against MenC. In School C, 5.56% of children were detected as vaccinated with only one dose of vaccine against MenA plus MenC, 59.26% vaccinated with two doses of vaccine against MenA plus MenC, 35.19% without a vaccination record of vaccine against MenA plus MenC and 59.26% vaccinated with vaccines against MenC (Table 4). According to the laboratory evidence and samples’ vaccination experience, three male MenC infection cases were further detected as breakthrough infection cases in addition to the index case. These three breakthrough cases were all studying in the eighth grade of School A while one of the them was a classmate and roommate of the index case.

## 4. Discussion

Vaccination is the best strategy to prevent MM and control meningitis outbreaks [17]. However, in this study, based on a large infection case that occurred in Tongzi County in southwestern China, some key issues were identified in the existing childhood MM immunization.

First, an enhanced emergency response can be made to protect high-risk populations through prophylactic medication and other measures. In this study, there were no other MM cases except for the index case and the results showed a low attack rate of MenC in School A. Therefore, it is necessary to point out other potential protective measures that may play roles in addition to vaccines; that is, 25 (17.2%) in 144 children surveyed from School A had prophylactic medication. When a higher proportion of the total population have prophylactic medication, the transmission route of meningitidis may be interrupted and those who do not have prophylactic medication may also benefit, resulting in higher effectiveness. Therefore, when the level of immunity is not sufficient to protect close contacts against some specific meningitidis serogroup, an emergency response in this way, together with social isolation measures, can be used to interrupt the main transmission route of the meningitidis in quite a short time [18]. However, it should be noted that mass prophylactic medications are always unfeasible and have a limited high cost and logistic problems [19]. Considering the common phenomenon of meningococcal-carrying among healthy children in this study, further consideration needs to be given to the potential contribution of vaccination in addition to prophylactic medication.

Second, there is a need to update the National Immunization Programme (NIP) to address the existing issues of meningitidis serogroups’ vaccines. Although no new secondary MM case was found in this study, there were still many children infected by healthy carriers. With regard to MenA and MenC, we found some infected children in this case, whereas MenA and MenC have been included in the NIP. Due to a paucity of systematical genomic evidence, we cannot speculate about the impact of exposure to a higher viral infection on breakthrough infections. What is certain is that the existing vaccines seem not to play a role in controlling transmission, which encourages vaccine producers to develop better vaccines or develop a booster schedule. It should also be noted that in the current NIP, two doses of the MenA polysaccharide vaccine should be administered to children at 6 and 9 months old and two doses of the MenA plus MenC polysaccharide vaccine should be administered to children at 3 years and 6 years, respectively. Further, children aged over 6 years old no longer receive boosting doses [20]. Therefore, for the children aged 9 to 15 years old included in this study, the protective effect could hardly persist [19,21,22]. It requires booster immunization in the NIP to prevent the high infection rate of MenA and MenC in older children, such as a quadratic meningococcal conjugate vaccine (MenACWY) for adults aged 11 or 12 years and a booster dose at age 16 years [23]. Some countries have expanded this vaccination schedule to older children. For example, Italy has extended the immunization population to those aged 18 years old in 2017, and Switzerland and Belgium opted for a vaccination strategy that included adolescents [24].

In addition to the MenA and MenC currently included in the NIP, we have also detected a large number of MenB-infected children. This is different from the existing literature evidence, perhaps due to the spatiotemporal population heterogeneity within China [22,25]. A national study has reported that Serogroup C, others and NG were the major reason among students aged over 7 years old but highlighted that one of the important tasks for MM control and prevention in the future is just to develop and provide new vaccines for Serogroup B [26]. In China, the MM cases caused by MenB have been rapidly increasing since 2015. The effectiveness of some MenB vaccines has reached 83%, and its protective effect can last for more than 3 years for 75% of the vaccinated [27,28]. However, neither 4CMenB nor rLP2086 (two mainstream MenB vaccines) have been introduced into the NIP [20], which may be another weakness of the existing immunization program. The active introduction and development of MenB-contained vaccines should be introduced to prevent the potential infection threat caused by MenB. Similar conclusions have been highlighted in other studies. For example, Truong [29] suggested after the introduction of a Hib vaccine, the leading cause of bacterial meningitis became *S pneumoniae*. The surveillance of potential changes in serotypes’ distribution over time is also encouraged, especially in low- and middle-income countries with limited resources for managing vaccine-preventable bacterial infectious diseases. In terms of China, the economic heterogeneity is large. Children are voluntary to vaccinate with the vaccines not included in the NIP, and those in less-developed regions are still at a great risk of being infected by the emerging leading serotype due to the low coverage of its corresponding vaccines. It should be noted that the distribution of pediatric bacterial meningitis causative organisms may vary by age [30], which should be also considered in the NIP.

Finally, although the children in School A showed the highest proportion of the population vaccinated against MenC, and the infection rates of MenC in School B and C were higher than in School A, the breakthrough infection and the attack after breakthrough infection both occurred only in this school. Further identification of the genotype of the C strain prevalent in the school should be carried out to explore the causes of the outbreak infection. In addition, distributions of different serogroup infection rates among schools showed significant differences. We tended to attribute these differences to different sources of infection among schools. On the one hand, although the only MenC case was in School A, a large number healthy carriers of other serogroups were still detected. On the other hand, School A and School B are relatively close and have close population linkage, making it reasonable to believe that their population is homogeneous. However, we still observed certain differences between Schools A and B, which helped us to exclude the possible impact of population heterogeneity on the distributions of serogroups. Therefore, genomic studies are necessary to be applied to detect the transmission pathway and further determine whether there are differences among the transmission ability of different serogroups, clarifying whether the existing vaccines are off target.

The contribution of this study is to provide the latest evidence to explore the weaknesses of the current MM outbreak control and prevention among school-age children in China, providing improvement strategies for the immunization program. On the one hand, it raised concerns about the effectiveness of the existing vaccines for older children and encouraged a wider immunization age period. On the other hand, it is suggested to actively introduce and develop corresponding vaccines based on the main meningitidis serogroups that cause MM cases in China to improve the NIP. From a global perspective, this study aimed to call on public health professionals from other parts of the world to pay attention to the attack and infection risk of older children. Also, this study hopes that the government will introduce more vaccines for adolescents and adults to avoid potential production losses.

This study has several limitations. First, this study only tested some main meningitis serogroups and cannot evaluate the infection of all serogroups. In the future, surveys targeting other meningitis serogroups should be encouraged to provide further evidence for understanding the full distribution of meningitis serogroups among Chinese children. Second, this study only carried out an exploratory analysis for the risk factors; thus, the conclusions of this study should be taken with caution. Due to the fact that the data in this study were from a survey that aims to detect the infection distribution in a public health emergency, more demographic characteristics have not been included in primary consideration. When more data are available, more causal inference techniques are encouraged to be applied here. Finally, due to the non-random sampling, the results may be influenced by the subjective judgment of the investigators. The representativeness of the results should be carefully viewed.

## 5. Conclusions

We evaluated the attack rates and the infection rates in different meningitidis serogroups of Neisseria meningitidis and provided strategies for meningitis outbreak controlling. On the one hand, effective emergency response measures could be considered as a short-term tool to prevent Neisseria meningitidis transmission. On the other hand, the extension of full-course immunization and active introduction and development of vaccines should be encouraged to improve the current NIP.

## Figures and Tables

**Table 1 vaccines-11-01762-t001:** The number and rate of infected cases in three schools.

	Number (Rate) of Infection Cases in Children in School A (*n* = 144)	Number (Rate) of Infection Cases in Children in School B (*n* = 68)	Number (Rate) of Infection Cases in Children in School C (*n* = 54)	Number (Rate) of Infection Cases in Total Children (*n* = 266)
Only MenA	11 (7.6%)	8 (11.8%)	6 (11.1%)	25 (9.4%)
Only MenB	2 (1.4%)	1 (1.5%)	5 (9.3%)	8 (3.0%)
Only MenC	9 (6.3%)	1 (1.5%)	1 (1.9%)	11 (4.1%)
MenA plus MenC	9 (6.3%)	21 (30.9%)	8 (14.8%)	38 (14.3%)
MenA plus MenB plus MenC	1 (0.7%)	1 (1.5%)	1 (1.9%)	3 (1.1%)
Only MenW	2 (1.4%)	NA	NA	NA
Unclassified Neisseria meningitides	1 (0.7%)	0	0	0

Note: School A is the school where the case was, School B is another middle school in the same town as School A, and School C is a middle school in a town far away from the town School A is in.

**Table 2 vaccines-11-01762-t002:** The number and rate of infection with different characteristics.

	Number (Rate) of Cases Infected with MenA	Number (Rate) of Cases Infected with MenB	Number (Rate) of Cases Infected with MenC	Number (Rate) of Cases Infected with MenW
**Gender**	*p* = 0.055	*p* = 0.397	*p* = 0.424	*p* = 0.187
Male	27 (19.9%)	7 (5.1%)	24 (17.6%)	2 (2.6%)
Female	39 (30.0%)	4 (3.1%)	28 (21.5%)	0
**School**	*p* < 0.001	*p* = 0.015	*p* = 0.002	NA
A	21 (14.6%)	3 (2.1%)	19 (13.2%)	2 (1.4%)
B	30 (44.1%)	2 (2.9%)	23 (33.8%)	NA
C	15 (27.8%)	6 (11.1%)	10 (18.5%)	NA
**Resident**	*p* = 0.320	*p* = 0.383	*p* = 0.851	*p* = 0.267
Day student	37 (27.4%)	7 (5.2%)	27 (20.0%)	0
Resident student	29 (22.1%)	4 (3.1%)	25 (19.1%)	2 (2.2%)

Note: School A is the school where the case was, School B is another middle school in the same town as School A, and School C is a middle school in a town far away from the town School A is in; Group A Neisseria meningitidis (MenA) includes only MenA, MenA plus MenC, and MenA plus MenB plus MenC; Group B Neisseria meningitidis (MenB) includes only MenB and MenA plus MenB plus MenC; and Group C Neisseria meningitidis (MenC) includes only MenC, MenA plus MenC, and MenA plus MenB plus MenC. Group W Neisseria meningitidis (MenW) denotes only MenW. NA: Laboratory tests on MenW were only conducted in school A. Therefore, number (rate) of cases infected with MenW are not applicable in school B and C.

**Table 3 vaccines-11-01762-t003:** Vaccination status in study objectives (region).

Region	MenA Plus MenC First Dose				MenA Plus MenC Boosting Dose			
	Expected Vaccination Number	Actual Vaccination Number	Unvaccination Number	Unvaccination Rate (%)	Expected Vaccination Number	Actual Vaccination Number	Unvaccination Number	Unvaccination Rate (%)
Zunyi City	1,293,686	1,285,160	8526	99.34	901,300	880,724	20,576	97.71
Tongzi County	124,018	123,147	871	99.29	83,186	80,931	2255	97.28
Liaoyuan Town	7154	7100	54	99.24	5219	5115	104	98

**Table 4 vaccines-11-01762-t004:** Vaccination status in study objectives (school).

School	The Proportion of Population Vaccinated Only One Dose of Vaccine against MenA Plus MenC (%)	The Proportion of Population Vaccinated Two Doses of Vaccine against MenA Plus MenC (%)	The Proportion of Population without Vaccination Record of Vaccine against MenA Plus MenC (%)	The Proportion of Population Vaccinated ACYW135 Vaccine (%)	The Proportion of Population Vaccinated Vaccine against MenC (%)
A	18.75	61.11	18.06	2.08	63.19
B	4.41	42.65	52.94	0	42.65
C	5.56	59.26	35.19	0	59.26

## Data Availability

The datasets used and/or analyzed during the current study are available from the corresponding author upon reasonable request.

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
