# Peer review of "What Works for Controlling Meningitis Outbreaks: A Case Study from China"

_vaccines, 2023, doi:10.3390/vaccines11121762_

Round 1

Reviewer 1 Report

Comments and Suggestions for Authors

The authors have made an interesting attempt at “What works for controlling meningitis outbreak, a case study from China”. The manuscript is interesting; however, the authors need to justify the scientific writing manuscript. Some of the general comments are provided below:

1.      How were the data collected and analyzed to understand the outbreak's dynamics? Could you provide more details on the immunization experience and testing conducted on students in the affected and other schools (A, B, C)?

2.     The article mentions "breakthrough infections" after full course immunization. What factors could contribute to breakthrough infections, and how might these impact vaccination strategies?

3.     Are there any ethical considerations regarding informed consent, privacy, or data sharing that were addressed during this investigation?

4.     The data presented show the distribution of meningococcal serogroups (MenA, MenB, MenC, MenW) among the infected children in schools A, B, and C. What are the implications of these differences in serogroup infection rates? Are there any known differences in the severity or outcomes associated with different serogroups?

5.     The mono-factor analysis highlights significantly different infection rates of MenC in school A, MenA in school B, and MenB in school C. What could explain these variations in infection rates among the schools?

6.     The vaccination rates among the children in Tongzi County are quite high, with over 97% receiving the second dose. Yet, breakthrough infections were still detected. What might be the reasons for breakthrough infections in this context, even with high vaccination coverage?

7.     The study highlights the need to update the National Immunization Programme (NIP) and address issues related to the coverage and age at which vaccines are administered. How can vaccination schedules be adapted to ensure long-term protection against meningitis, especially for older children? Are there international best practices for age-specific vaccination?

8.     The study mentions spatiotemporal population heterogeneity within China as a factor contributing to variations in serogroup infections. How can public health efforts be tailored to address these regional differences? What challenges exist in implementing consistent vaccination strategies across diverse populations?

9.     How might the findings of this study apply to other regions with varying demographics and healthcare systems? Are there lessons that can be learned and adapted for use in different parts of the world?

Author Response

Comments from Reviewer 1:

We would like to express our sincere thanks to the reviewer for the constructive and positive comments. The main corrections in the paper and the responds to the reviewers’ comments are as follows:

  1. How were the data collected and analyzed to understand the outbreak's dynamics? Could you provide more details on the immunization experience and testing conducted on students in the affected and other schools (A, B, C)?

Response: We thank the reviewer for this helpful comment. This study is not a follow-up survey, but a cross-sectional survey based on CDC post-hoc emergency response survey. The first case (i.e., index case) was reported by the hospital through the China Notifiable Disease Reporting System. Subsequently, the provincial, municipal and county CDC carried out an active search for suspected meningitis cases in the region the incident case located in, and daily surveillance for suspected meningitis cases in the school the incident case located in until 10 days after the index has left the school (the longest incubation period). No other meningitis was found except for the index case.

The immunization experience of children in this study were surveyed by checking vaccination certificates, vaccination cards and parents' memories. A total of 266 children were surveyed, including 144 in School A, 68 in School B and 54 in School C. Under informed consent, signing an informed consent statement, surveyors were allowed to collect venous blood and detect meningitis antibodies from children. Blood of a total of 246 children were collected, including 124 in School A, 68 in School B and 54 in School C.

We have added this description on the Data collection and analysis section (highlighted in red in the manuscript), as follows:

“The immunization experience of children in this study were surveyed by checking vaccination certificates, vaccination cards and parents' memories. A total of 266 children were surveyed, including 144 in School A, 68 in School B and 54 in School C. Under in-formed consent, signing an informed consent statement, surveyors were allowed to collect venous blood and detect meningitis antibodies from children. Blood of a total of 246 children were collected, including 124 in School A, 68 in School B and 54 in School C.”

  1. The article mentions "breakthrough infections" after full course immunization. What factors could contribute to breakthrough infections, and how might these impact vaccination strategies?

Response: We thank the reviewer for raising this point. Your comment has helped us further summarize the potential policy implications of our study. We have summarized the factors could contribute to breakthrough infections and the impact of breakthrough infections on immune strategies.

The factors could contribute to breakthrough infections

Factors

Mechanism

Reference

Time

While the amount of circulating antibody present following vaccination (or any antigenic stimulus) increases rapidly, on a timescale of days to weeks, it also declines rapidly from its peak on a timescale of weeks to months, and then more slowly over a time scale of decades.

[1,2]

Immunodeficient individuals

For example, older individuals, whose neutralizing antibody responses to COVID-19 vaccines are typically lower, appear to be at greater risk of breakthrough infections at any given time following vaccination.

[3,4]

Exposure to a higher viral inoculum

Exposure to a higher viral inoculum can reduce vaccine effectiveness and increase the probability of breakthrough infection.

[5,6]

The impact of breakthrough infections on immune strategies

Your comment helped us fulfill the importance of focusing on breakthrough infections. Identifying the frequency, severity and causes of breakthrough infections may inform the choice of public health responses: watchful waiting may be appropriate if breakthroughs are comparatively rare or mild and unlikely to markedly increase transmission rates. By contrast, if breakthrough infections are common, severe or highly transmissible, then there may be a need for additional vaccine doses, changes in vaccine formulations or non-pharmaceutical interventions (or a combination of these approaches) to reduce the incidence of infection[6].

We have added this description on the Introduction section (highlighted in red in the manuscript), as follows:

“In addition, the decline of antibody induced by vaccination, immuno-deficient individuals and exposure to a higher viral inoculum may cause breakthrough infections, which has potential impact on the vaccination strategies. On one hand, if the breakthrough infections occurred rarely or mildly and have comparatively low probability to cause wide transmission, watchful waiting may be appropriate. On the other hand, if the breakthrough infections are comparatively common, additional vaccine doses, changes in vaccine formulations or non-pharmaceutical interventions should be considered as response.”

Reference

  1. Levin, E.G.; Lustig, Y.; Cohen, C.; Fluss, R.; Indenbaum, V.; Amit, S.; Doolman, R.; Asraf, K.; Mendelson, E.; Ziv, A.; et al. Waning Immune Humoral Response to BNT162b2 Covid-19 Vaccine over 6 Months. N Engl J Med 2021, 385, e84, doi:10.1056/NEJMoa2114583.
  2. Amanna, I.J. Duration of Humoral Immunity to Common Viral and Vaccine Antigens. The New England Journal of Medicine 2007.
  3. Lustig, Y.; Sapir, E.; Regev-Yochay, G.; Cohen, C.; Fluss, R.; Olmer, L.; Indenbaum, V.; Mandelboim, M.; Doolman, R.; Amit, S.; et al. BNT162b2 COVID-19 Vaccine and Correlates of Humoral Immune Responses and Dynamics: A Prospective, Single-Centre, Longitudinal Cohort Study in Health-Care Workers. The Lancet Respiratory Medicine 2021, 9, 999–1009, doi:10.1016/S2213-2600(21)00220-4.
  4. Scobie, H.M.; Johnson, A.G.; Suthar, A.B.; Severson, R.; Alden, N.B.; Balter, S.; Bertolino, D.; Blythe, D.; Brady, S.; Cadwell, B.; et al. Monitoring Incidence of COVID-19 Cases, Hospitalizations, and Deaths, by Vaccination Status — 13 U.S. Jurisdictions, April 4–July 17, 2021. MMWR Morb. Mortal. Wkly. Rep. 2021, 70, 1284–1290, doi:10.15585/mmwr.mm7037e1.
  5. Gomes, M.G.M.; Lipsitch, M.; Wargo, A.R.; Kurath, G.; Rebelo, C.; Medley, G.F.; Coutinho, A. A Missing Dimension in Measures of Vaccination Impacts. PLoS Pathog 2014, 10, e1003849, doi:10.1371/journal.ppat.1003849.
  6. Lipsitch, M.; Krammer, F.; Regev-Yochay, G.; Lustig, Y.; Balicer, R.D. SARS-CoV-2 Breakthrough Infections in Vaccinated Individuals: Measurement, Causes and Impact. Nat Rev Immunol 2022, 22, 57–65, doi:10.1038/s41577-021-00662-4.
  7. Are there any ethical considerations regarding informed consent, privacy, or data sharing that were addressed during this investigation?

Response: We thank the reviewer for raising this point. Our data collection and analysis have been ethically reviewed. The children under investigation under informed consent and signed an informed consent statement. The raw data is processed and analyzed by the Guizhou Center for Disease Control and Prevention and properly kept.

“ The study was conducted in accordance with the Declaration of Helsinki, and approved by the Institutional Review Board of Guizhou Center for Disease Control and Prevention (protocol code Q2023-10).”

“ All participants completed and signed a consent form approved by the Institutional Review Board of Guizhou Center for Disease Control and Prevention.”

  1. The data presented show the distribution of meningococcal serogroups (MenA, MenB, MenC, MenW) among the infected children in schools A, B, and C. What are the implications of these differences in serogroup infection rates? Are there any known differences in the severity or outcomes associated with different serogroups?

Response: Many thanks for this helpful suggestion.

First, these differences in serogroup infection rates contributed to trace the source. As what we replied in Question 5, we attributed these differences to different sources of infection among schools. In the future, genomic studies can be applied to detect the transmission pathway and further determine whether there are differences among the transmission ability of different serogroups, clarifying whether existing vaccines are off target. Our study provided a rudimentary understanding, but still helps to clarify the direction of further in-depth studies and provide a basis for policymaking.

Second, there is no other meningitis incident in this study in addition to the only index case. Although meaningful and interesting, we have not detected any known differences in the severity or outcomes associated with different serogroups.

We have added this description on the Discussion section (highlighted in red in the manuscript), as follows:

“In addition, distributions of different serogroup infection rates among schools showed significant difference. We tend to attribute these differences to different sources of infection among schools. On one hand, although the only MenC case was in school A, a large number healthy carriers of other serogroups were still detected. On another hand, School A and School B are relatively close and have close population linkage, making it reasonable to believe that their population is homogeneous. However, we still observed certain differences between schools A and B, which helped us excluding the possible impact of population heterogeneity on the distributions of serogroups. Therefore, genomic studies are necessary to be applied to detect the transmission pathway and further determine whether there are differences among the transmission ability of different serogroups, clarifying whether existing vaccines are off target.”

  1. The mono-factor analysis highlights significantly different infection rates of MenC in school A, MenA in school B, and MenB in school C. What could explain these variations in infection rates among the schools?

Response: We thank the reviewer for pointing out this ambiguity.

We believed that population susceptibility heterogeneity (e.g, enrollment bias), different sources of infection, and differences in immunization experience may be factors affected.

First, since the meningitis A+C vaccine has been included in NIP in China, there is almost no difference in children's immunization experience.

Second, population susceptibility heterogeneity is also excluded. School A and School B are geographically close and have close population linkage, making it reasonable to believe that their population is homogeneous. However, we still observed certain differences between schools A and B, which helped us excluding the possible impact of population heterogeneity on the distributions of serogroups.

Third, although the only MenC case was in school A, a large number of healthy carriers of other serogroups were still detected. Therefore, we preferred to tend to attributed these differences to different sources of infection among schools.

We added this description on the Discussion section (highlighted in red in the manuscript), as follows:

“ In addition, distributions of different serogroup infection rates among schools showed significant difference. We tended to attribute these differences to different sources of infection among schools. On one hand, although the only MenC case was in school A, a large number healthy carriers of other serogroups were still detected. On another hand, School A and School B are relatively close and have close population linkage, making it reasonable to believe that their population is homogeneous. However, we still observed certain differences between schools A and B, which helped us exclude the possible impact of population heterogeneity on the distributions of serogroups. Therefore, genomic studies are necessary to be applied to detect the transmission pathway and further determine whether there are differences among the transmission ability of different serogroups, clarifying whether existing vaccines are off target.”

  1. The vaccination rates among the children in Tongzi County are quite high, with over 97% receiving the second dose. Yet, breakthrough infections were still detected. What might be the reasons for breakthrough infectionsin this context, even with high vaccination coverage?

Response: We thank the reviewer for this comment. As answered in Question 2, time, immunodeficient individuals, and exposure to a higher viral infection are the main potential factors for breakthrough infections. The older children in this study are not known since immunodeficiency is rare, therefore, we believed the insufficient of antibody may be main contributor to the breakthrough.

In fact, due to a paucity of systematical genomic evidence, we cannot be certain about the impact of exposure to a higher viral infection on breakthrough infections. Based on such considerations, we did not exclude the impact of exposure to a higher virtual inventory in the Discussion section, as follows:

“Due to a paucity of systematical genomic evidence, we cannot be speculated about the impact of exposure to a higher viral infection on breakthrough infections. What can be certain is that existing vaccines seem not play a role in controlling transmission, which encourages vaccine producers to develop better vaccines or develop a booster schedule.”

In terms of time, there are some course evidences to prove it played an important role. For example, almost the final dose of meningitis vaccines for all the children in this study had been over 5 years until the index case was reported. Therefore, the protective effect could hardly persist[1–3].

“It should also be noted that in current NIP, two doses of MenA polysaccharide vaccine should be administered to children at 6 and 9 months old and two doses of MenA plus MenC polysaccharide vaccine should be administered to children at 3 years and 6 years respectively. Further, children aged over 6 years old no longer receive boosting doses. Therefore, for the children aged 9 to 15 years old included in this study, the protective effect could hardly persist.”

Reference

  1. Butt, A.A.; Khan, T.; Yan, P.; Shaikh, O.S.; Omer, S.B.; Mayr, F. Rate and Risk Factors for Breakthrough SARS-CoV-2 Infection after Vaccination. Journal of Infection 2021, 83, 237–279, doi:10.1016/j.jinf.2021.05.021.
  2. Memish, Z.A. Meningococcal Disease and Travel.
  3. Zhang, Y.; Wei, D.; Guo, X.; Han, M.; Yuan, L.; Kyaw, M.H. Burden of Neisseria Meningitidis Infections in China: A Systematic Review and Meta–Analysis. Journal of Global Health 2016, 6, 020409, doi:10.7189/jogh.06.020409.
  4. The study highlights the need to update the National Immunization Programme (NIP) and address issues related to the coverage and age at which vaccines are administered. How can vaccination schedules be adapted to ensure long-term protection against meningitis, especially for older children? Are there international best practices for age-specific vaccination?

Response: We thank the reviewer for raising these helpful suggestions.

We suggest extending NIP and incorporating more booster vaccines suitable for older children and adults to ensure long-term protection against meningitis. Strengthen neutralizing anti body responses through subsequent immune responses. According to the Advisory Committee on Immunization Practices[1], it recommends routing vaccination with a quadratic meningococcal conjugate vaccine (Men ACWY) for adults aged 11 or 12 years, with a boost dose at age 16 years. Improved immunization plans can help improve the risk of infections and attacks in school-age children. Corresponding revision in the manuscript is as shown:

“It requires booster immunization in NIP to prevent the high infection rate of MenA and MenC in older children, such as a quadratic meningococcal conjugate vaccine (MenACWY) for adults aged 11 or 12 years and a boost dose at age 16 years.”

Reference

  1. Centers for Disease Control and Prevention; Prevention Prevention and Control of Meningococcal Disease: Recommendations of the Advisory Committee on Immunization Practices (ACIP). In Pediatric Clinical Practice Guidelines & Policies; American Academy of Pediatrics, 2014; pp. 1101–1102 ISBN 978-1-58110-861-3.
  2. The study mentions spatiotemporal population heterogeneity within China as a factor contributing to variations in serogroup infections. How can public health efforts be tailored to address these regional differences? What challenges exist in implementing consistent vaccination strategies across diverse populations?

Response: Region-specific vaccination strategies can help address these regional differences, but their implementation requires regional evidence as support. Unfortunately, our study is not sufficient to detect the differences within China. Some studies implicated us with regional differences within China[1,2] and even the world[3]. We hope that these differences can be well discussed in more national surveys, systematic reviews, and meta-analyses in the future, and further served as policymaking basis. In addition, the governance capacity of the regional healthcare sectors should also be evaluated in advance, as these region-specific policies will pose higher challenges to them.

We have added references followed on the Discussion section:

  1. Zhang, Y.; Wei, D.; Guo, X.; Han, M.; Yuan, L.; Kyaw, M.H. Burden of Neisseria Meningitidis Infections in China: A Systematic Review and Meta–Analysis. Journal of Global Health 2016, 6, 020409, doi:10.7189/jogh.06.020409.
  2. Wang, B.; Lin, W.; Qian, C.; Zhang, Y.; Zhao, G.; Wang, W.; Zhang, T. Disease Burden of Meningitis Caused by Streptococcus Pneumoniae Among Under-Fives in China: A Systematic Review and Meta-Analysis. Infect Dis Ther 2023, doi:10.1007/s40121-023-00878-y.
  3. Oordt-Speets, A.M.; Bolijn, R.; Van Hoorn, R.C.; Bhavsar, A.; Kyaw, M.H. Global Etiology of Bacterial Meningitis: A Systematic Review and Meta-Analysis. PLoS ONE 2018, 13, e0198772, doi:10.1371/journal.pone.0198772.
  4. How might the findings of this study apply to other regions with varying demographics and healthcare systems? Are there lessons that can be learned and adapted for use in different parts of the world?

Response: We thank the reviewer for pointing out this concern.

This study filled the gap in previous studies on older children and identifies the risks of infections and attacks faced by older children in schools. It also suggested that the current high vaccination rate cannot completely eliminate the future meningitis risks.

On the one hand, we hope to call on public health professionals from other parts of the world to pay attention to older children and their potential risks of infections and attacks in school through our findings. Even in other regions with varying demographics and healthcare systems, older children may be facing similar problems. These prevailing conditions are masked by the attacks and infections of infants and young children and rarely discussed.

On the other hand, we also hope that in the future, under economic availability, the government will involve more vaccines for adolescents and adults. Infectious diseases also require attention in the lifecycle stages after infants and young children, as their morbidity and mortality may also reduce the potential gross domestic product of the society. We have supplemented the contents below on the Discussion section.

“ From a global perspective, this study aimed to call on public health professionals from other parts of the world to pay attention to the attack and infection risk of older children. Also, this study hopes that government will introduce more vaccines for adolescents and adults to avoid potential production losses.”

Reviewer 2 Report

Comments and Suggestions for Authors

Estimated authors,

I've read with great interest the present narrative review, reporting on the topic of breakthrough infections following pneumococcal vaccination. The paper is well written and organized, as well as well documented. From my point of view, there are only two minor shortcomings that could be amended by minor adjustments of the main text:

1) provide early across the main text a working definition of breakthrough infection;

2) similarly, provide within the main text (I would say, in the first sections) some and concise informations about the differences between PCV and PPSV, in order to highlight as better as possible which characteristics of the former represent a significant improvement over the latter.

Author Response

Comments from Reviewer 2:

We would like to express our sincere thanks to the reviewer for the constructive and positive comments. The main corrections in the paper and the responds to the reviewers’ comments are as follows:

  1. provide early across the main text a working definition of breakthrough infection.

Response: Many thanks for this helpful suggestion. Following this suggestion, we have added the definition on the Introduction section. They read as follows:

“In addition, breakthrough infection, defined that throat swabs are detected neisseria meningitidis carriers after full course immunization with the corresponding meningitis serogroup vaccine, is another huge challenge for immunization plans.”

  1. similarly, provide within the main text (I would say, in the first sections) some and concise informations about the differences between PCV and PPSV, in order to highlight as better as possible which characteristics of the former represent a significant improvement over the latter.

Response: We thank the reviewer for pointing out this ambiguity, followed by your suggestion, we added a detailed description of current national immunization plan and the difference between PCV and PSV.

“In addition, vaccine characteristics also matters. At present, there are 5 types of meningococcal meningitis vaccines in China, with MPSV-A and MPSV-AC involved in the national immunization plan, and they are vaccinated free of charge at the sixth month, ninth month, third and sixth year after birth, respectively. The remaining MPCV-AC, MPSV-ACYW135, and MPCV-ACYW135 need to be vaccinated by eligible children at their own expense. Therefore, most infantes and children vaccinated with PSV. However, PSV only produces a brief immune response among infants and young children under 2 years, without the immune memory enhancement effect of repeated vaccination; while PCV vaccine not only produces a good immune response among infants and young children under 2 years, but also has a memory enhancing immune effect after repeated vaccination. This prevailing condition may cause the concerns on the long-term protection of meningitis.”

Reviewer 3 Report

Comments and Suggestions for Authors

Thank you for sharing your manuscript on meningococcal meningitis. Here some suggested comments that could help to improve the article:

Introduction:

-Please be more specific on the time frame, i.e., "recent years" when the incidence ranged from 0.07-12.6 cases/100,000 population

-Please also be more specific on the time frame when an incidence of 0.09 cases/100,000 population was found in China.

-Please include more specific data on mortality seen in China rather than giving a general statement.

-Please state the actual vaccine administered including the mode and schedule of vaccine administration as well as the serogroups covered that resulted in 90% MM reduction.

-When reporting MM among Chinese children also in day-care facilities and schools, please state their age range. 

-Please be more specific on the so called "brief" immune response among infants and young children below 2 years of age reported for PSV. The same applies to PCV in terms of the so called "good immune response" among the same age group. 

Method:

-Please include in your manuscript the laboratory testing performed on the case of November 16/17, 2022.

-How do you define an index case within the context of your research? 

-Please include more information on the re-vaccination of individuals aged below 18 years using the MenA plus MenC vaccine. Also, please provide more information on the treatment of suspected MM cases. 

-Please include more information how the screening/search for additional MM cases was performed. As no new cases were detected, maybe the so called index case actually was not the index case?

-How were surveyed children selected to assure randomness of your sample? What inclusion/exclusion criteria did participants have to meet? What age did the participating children have? Did you also obtain the consent of parents/caregiver in case of minors? 

Comments on the Quality of English Language

Please see above.

Author Response

Comments from Reviewer 3:

We would like to express our sincere thanks to the reviewer for the constructive and positive comments. The main corrections in the paper and the responds to the reviewers’ comments are as follows:

  1. Please be more specific on the time frame, i.e., "recent years" when the incidence ranged from 0.07-12.6 cases/100,000 population

Response: We thank the reviewer for pointing out this ambiguity. Followed by your suggestion, we rechecked the cited reference and replaced “in recent years” with “from 1950–2002” (page 2 in red).

  1. Please also be more specific on the time frame when an incidence of 0.09 cases/100,000 population was found in China.

Response: We thank the reviewer for pointing out this ambiguity. Related data is from an article titled “Prevalence of meningococcal meningitis in China from 2005 to 2010”, an incidence of meningitis from 2005 to 2010 is 0.09 cases/100,000 population. We have added it as a new reference and also added this time period into the corresponding sentence as follows:

“The incidence of MM reached the peak in the spring in 1967 with 403 cases per 100,000 population in China, whereas has been controlled at 0.09 cases per 100,000 population from 2005 to 2010 contributing to governments’ long-term efforts on vaccines development and immunization[1] (page 2 in red).”

Reference

[1] Li, J.; Li, Y.; Shao, Z.; Li, L.; Yin, Z.; Ning, G.; Xu, L.; Luo, H. Prevalence of Meningococcal Meningitis in China from 2005 to 2010. Vaccine 2015, 33, 1092–1097, doi:10.1016/j.vaccine.2014.10.072

  1. Please include more specific data on mortality seen in China rather than giving a general statement.

Response: Many thanks for this helpful suggestion. Following this suggestion, we have added the specific data on mortality in China on the Introduction section. They read as follows:

“Compared with the mortality rate of 0.30 per 100,000 population in United States of America in 2019, the mortality in China has reached 0.51 per 100,000 population ac-cording to GBD 2019 study  (page 2 in red) [1].”

Reference

[1] Vos, T.; Lim, S.S.; Abbafati, C.; Abbas, K.M.; Abbasi, M.; Abbasifard, M.; Abbasi-Kangevari, M.; Abbastabar, H.; Abd-Allah, F.; Abdelalim, A.; et al. Global Burden of 369 Diseases and Injuries in 204 Countries and Territories, 1990–2019: A Systemat-ic Analysis for the Global Burden of Disease Study 2019. The Lancet 2020, 396, 1204–1222, doi:10.1016/S0140-6736(20)30925-9.

  1. Please state the actual vaccine administered including the mode and schedule of vaccine administration as well as the serogroups covered that resulted in 90% MM reduction.

Response: We thank the reviewer for raising this point. Your comment has helped us improve the description on the importance of vaccine introduction on burden reduction. We have revised the related sentence as follows:

“The introduction and utilization of conjugate meningococcal C vaccines, in the UK and the USA, which uses a 2, 4, 6, and 12–15 month schedule, reduced N meningitidis serogroup C disease by over 90%, and near elimination of Haemophilus influenzae has been documented due to the introduction of conjugate Hib vaccines (page 2 in red).”

  1. When reporting MM among Chinese children also in day-care facilities and schools, please state their age range. 

Response: Many thanks for this helpful suggestion. We have added the age range of Chinese children also in day-care facilities as follows:

“The outbreak of MM was especially more frequently to detect in children. In day-care settings or schools, healthy children (aged 6-18 years old in China) can be infected under close contact with MM cases or asymptomatic carriers, which lead to further outbreaks (page 2 in red).”

  1. Please be more specific on the so called "brief" immune response among infants and young children below 2 years of age reported for PSV. The same applies to PCV in terms of the so called "good immune response" among the same age group. 

Response: We thank the reviewer for pointing out this ambiguity. We have rechecked the literature and corrected the ambiguous description. Actually, nonconjugated pneumococcal polysaccharide vaccines do not elicit a protective immune response in children younger than 2 years [1]. Also, CDC (https://www.cdc.gov/vaccines/vpd/pneumo/public/index.html#how-well-vaccines-work) pointed out “Children younger than 2 years old should not get PPSV23.” Revised sentence in the manuscript reads as follows:

“However, on one hand, nonconjugated pneumococcal polysaccharide vaccines do not elicit a protective immune response in children younger than 2 years[1] and only PCV vaccine has a memory enhancing immune effect after repeated vaccination (page 3 in red).”

Reference

[1] Dagan, R. Relationship between Immune Response to Pneumococcal Conjugate Vaccines in Infants and Indirect Protection af-ter Vaccine Implementation. Expert Review of Vaccines 2019, 18, 641–661, doi:10.1080/14760584.2019.1627207.

  1. Please include in your manuscript the laboratory testing performed on the case of November 16/17, 2022.

Response: We thank the reviewer for this helpful comment. We have added the detail on the laboratory testing performed on the case of November 16/17, 2022 as follows:

“In the afternoon on November 16, the patient was transferred to a higher-Grade hospital and diagnosed as “Suspected MM case” and was uploaded to the pandemic network at 8:38 on November 17, and further confirmed as meningitidis serogroup C by laboratory diagnosis. The main method is to use real-time PCR to detect Neisseria meningitidis species and specific nucleic acid fragments of common serogroups in blood samples, and to capture specific genes of Neisseria meningitidis species (CtrA gene and Group C serogroup specific gene). The patient died on November 20 and was eventually defined as the index case (page 4 in red).

  1. How do you define an index case within the context of your research? 

Response: We thank the reviewer for raising this point. An index case refers to the case that met the case definition during an outbreak and was first detected and reported. We employed this definition to provide clues for tracking the transmission of pathogens, identify the causes of outbreaks, and further propose control measures. We have added this description on the Methods section, as follows:

“An index case refers to the case that infected with pathogens during an infection outbreak and was first detected and reported (page 4 in red).”

  1. Please include more information on the re-vaccination of individuals aged below 18 years using the MenA plus MenC vaccine. Also, please provide more information on the treatment of suspected MM cases. 

Response: We thank the reviewer for raising this point. Following your suggestion, we have added more information on the re-vaccination. What should be noticed is that there was no suspected MM cases after active search due to that no suspicious symptoms such as fever, headache, and vomiting were found. We have revised as follows:

“Second, they ordered all medical institutions and townships to carry out active search for suspected cases, improve awareness of suspicious symptoms of MM, report suspicious cases and carry out isolation and treatment in a timely manner. Active search for additional MM cases was performed through symptom monitoring in schools, communities and Hospital Information System. Specifically, six carriers were detected among index case’s close contacts at the same school. These carriers were isolated and observed at home for 10 days, and no suspicious clinical symptoms found. Medical observation was also conducted on their family members of the index case and carriers above, as well as the school's teachers and students, and no suspicious symptoms, such as fever, headache, and vomiting, were found. There was no suspected MM case and subsequent incident in this outbreak infection. Finally, the local government encouraged residents aged under 18 to carry out doses MenA plus MenC vaccine revaccination for those unvaccinated before. For the school where the index case is located, the immun-ization experience of students was verified through systematic verification combined with child vaccination certificates. For other residents aged over 24 months having not com-plete vaccination against MM in the past, residents were vaccinated with doses MenA plus MenC vaccine according to the immunization program. Residents can also choose ACYW135 vaccines (not included in current immunization programme) as an alternative to vaccinate based on the principles of being informed, voluntary, and self-funded (page 4-5 in red).

  1. Please include more information how the screening/search for additional MM cases was performed. As no new cases were detected, maybe the so called index case actually was not the index case?

Response: We thank the reviewer for this helpful comment. Active search for additional MM cases was performed through symptom monitoring in schools, communities and Hospital Information System. Corresponding description has been added in to the manuscript, as followed:

“Second, they ordered all medical institutions and townships to carry out active search for suspected cases, improve awareness of suspicious symptoms of MM, report suspicious cases and carry out isolation and treatment in a timely manner. Active search for additional MM cases was performed through symptom monitoring in schools, communities and Hospital Information System. Specifically, six carriers were detected among index case’s close contacts at the same school. These carriers were isolated and observed at home for 10 days, and no suspicious clinical symptoms found. Medical observation was also conducted on their family members of the index case and carriers above, as well as the school's teachers and students, and no suspicious symptoms, such as fever, headache, and vomiting, were found. There was no subse-quent incident of this outbreak infection. (page 4 in red)”

In addition, based on your previous questions about the definition of index case, we have made corresponding supplements. It should be noted that in an infection outbreak, we just define an index case where the interest lies in the infected person firstly detected and reported, rather than the incident. Thank you again for your question about the index case, which has indeed helped us correct the previously ambiguous expression.

  1. How were surveyed children selected to assure randomness of your sample? What inclusion/exclusion criteria did participants have to meet? What age did the participating children have? Did you also obtain the consent of parents/caregiver in case of minors? 

Response:  Thank you for your insight comment. We have ignored the description of sampling in previous versions of manuscript.

This study is an emergency public health survey aiming at quickly identifying the transmission routes and controlling the continued spread of pathogens. Therefore, this study adopted judgment sampling based on the professional knowledge of CDC investigators. A certain number of students aged 10-15 in the same dormitory, class, grade, school, and the same age group from other schools with the index case were surveyed. Due to the non-random sampling, the results may be influenced by the subjective judgment of the investigators. We should remind readers to carefully view the representativeness of the results. Therefore, in addition to supplementing corresponding descriptions in Methods section, we have also revised limitations as follows:

“This study adopted judgment sampling based on the professional knowledge of CDC investigators. A certain number of students aged 10-15 in the same dormitory, class, grade, school, and the same age group from other schools with the index case were surveyed (page 5 in red).”

“Finally, due to the non-random sampling, the results may be influenced by the subjective judgment of the investigators. The representativeness of the results should be carefully viewed (page 12 in red).”

As the samples are all resident students, the on-site investigation was conducted with the consent and assistance of the school caregiver, and the information inquiry was conducted with the consent and support of the student's family caregiver. Corresponding revision reads as followed:

“On-site survey was conducted with the consent and assistance of the school, and the information inquiry was conducted with the consent and support of the student's family caregiver (page 13 in red).”

Round 2

Reviewer 1 Report

Comments and Suggestions for Authors

The authors have addressed my questions, and now the manuscript is acceptable for publication. 

Author Response

Special thanks to you for your good comments.

Reviewer 3 Report

Comments and Suggestions for Authors

Thank you for addressing all my comments sufficiently. 

Comments on the Quality of English Language

Please see above.